# CogView2: Faster and Better Text-to-Image Generation via Hierarchical Transformers

**Ming Ding**[†]  **Wendi Zheng**[†]  **Wenyi Hong**[†]  **Jie Tang**[†‡]
[†]Tsinghua University  [‡]BAAI
{dm18@mails, jietang@mail}.tsinghua.edu.cn

## Abstract

Development of transformer-based text-to-image models is impeded by its slow generation and complexity, for high-resolution images. In this work, we put forward a solution based on hierarchical transformers and local parallel autoregressive generation. We pretrain a 6B-parameter transformer with a simple and flexible self-supervised task, a cross-modal general language model (CogLM), and fine-tune it for fast super-resolution. The new text-to-image system, CogView2, shows competitive generation performance to the concurrent state-of-the-art DALL-E-2, and naturally supports interactive text-guided editing on images.

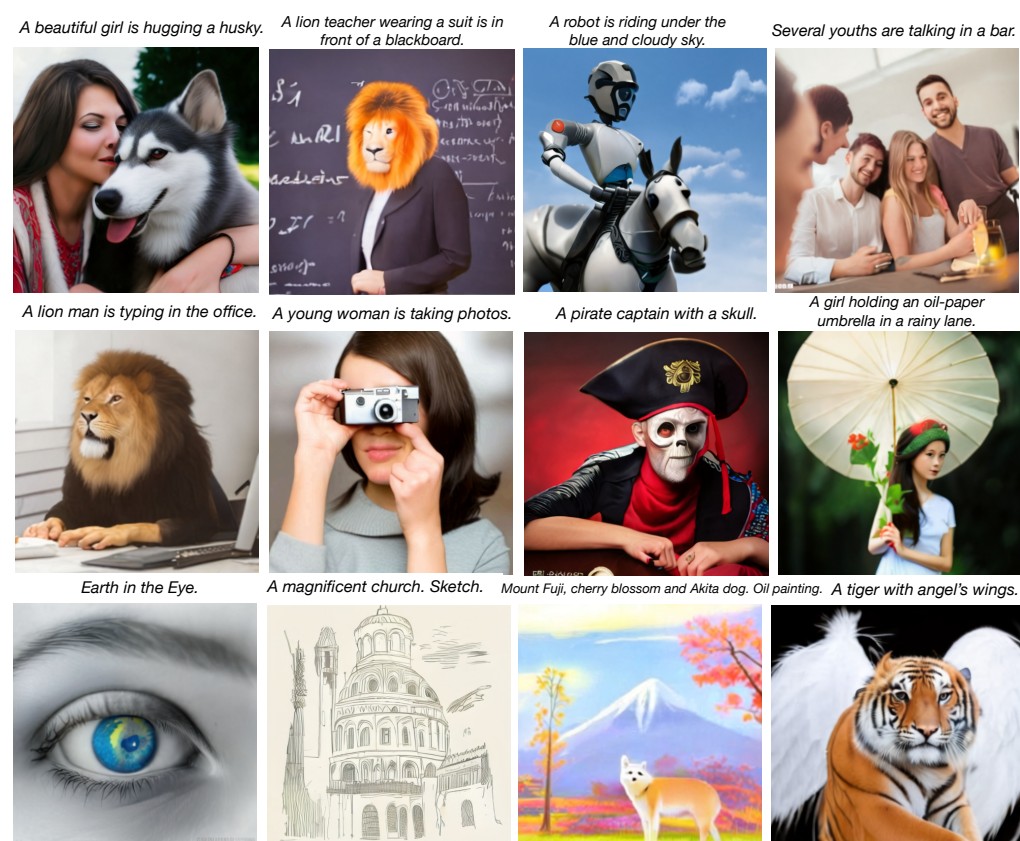

Figure 1: Text-to-Image samples from CogView2, which supports **both Chinese and English**. The actual input text is in Chinese, translated into English here for better understanding. Codes and a demo website will be updated at https://github.com/THUDM/CogView2.

36th Conference on Neural Information Processing Systems (NeurIPS 2022).

# 1 Introduction

Recently, text-to-image generation has been greatly advanced by large-scale pretrained transformers, e.g. DALL-E [26] and CogView [3]. These models learn to generate image tokens in an autoregressive way. However, they also suffer from the following disadvantages:

**Slow generation.** Generation of autoregressive models usually is much slower than generation of non-autoregressive models, e.g. GANs [10], with the same FLOPs. Instead of employing a large number of parameters, this shortcoming is mainly attributed to the nature of token-by-token generation used in the autoregressive models cannot exploit the parallel computing ability of GPUs, even after caching hidden states [25]. This is a significant limitation.

**Expensive high-resolution training.** The current large-scale pretrained models are generally based on Transformers [30], where the attention operation has both time and space complexity of $O(n^2)$ for training sequences of length $n$. Within a limited budget, we face a trade-off between the number of parameters, representing the modeling power, and the resolution of the generated images. For this reason, most current text-to-image models choose a resolution of $32 \times 32$ tokens (usually $256 \times 256$ pixels) [3, 26, 11], which is far less dense than the resolution of the real photos.

**Unidirectionality.** For images, autoregressive models, e.g. GPTs, usually generate tokens in raster-scan order. This order shows the best perplexity during the evaluation [7]. However, this order makes the models unaware of the tokens below or on the right side during generation, as a result text-guided *infilling* is not supported. Moreover, the unidirectionality leads to a gap between the pretrained text-to-image models and vision transformers (ViTs) [5] based on bidirectional masked prediction, e.g. MAE [12] and SimMIM [34]—limiting their application on traditional visual tasks, such as image classification and object detection.

**Present Work.** To overcome these defects, we first propose a simple and versatile pretraining method, a **C**ross-**M**odal **g**eneral **L**anguage **M**odel (CogLM). Our CogLM masks various types of tokens in the sequence of text and image tokens, and learns to predict them autoregressively. Specifically, (1) if we mask all the image tokens, the task becomes the same as the original CogView [3] in performing a text-to-image generation task; (2) if we mask random patches of image tokens, it works similarly to MAE as an infilling task; (3) if we mask text tokens, the task becomes image captioning.

The versatility of CogLM enables us to fine-tune a pretrained CogLM for different downstream tasks, and constructs a hierarchical model, CogView2.There are three steps in the hierarchical generation process as follows:

1. First, we generate a batch of low-resolution images ($20 \times 20$ tokens in CogView2) using the pretrained CogLM, and then (optionally) filter out the bad samples based on the perplexity of CogLM image captioning, which is the post-selection method introduced in CogView [3].

2. The generated images are mapped into $60 \times 60$-token images by a *direct super-resolution* module fine-tuned from the pretrained CogLM. We use local attention implemented by our customized CUDA kernel to reduce the training expense. The high-resolution images from this step usually have inconsistent textures and lack details.

3. These high-resolution images are refined via another *iterative super-resolution* module fine-tuned from the pretrained CogLM. Most tokens are re-masked and re-generated in a *local parallel autoregressive* (LoPAR) way, which is much faster than the original autoregressive generation.

How does CogView2 conquer the three defects? First, during pretraining the masked patch prediction task trains CogLM to handle bidirectional context, making it easy to adapt to bidirectional tasks, such as direct and iterative super-resolution. Second, the hierarchical design allows us to care only about the local coherence at a high-resolution level. In this way, the local attention can be leveraged to reduce the training expense. Third, the local parallel autoregressive generation can reduce model run times from 3,600 to 6 (1/600 only), significantly accelerating the generation of high-resolution images. CogView2 is about $10\times$ faster than the CogView (with sliding-window super-resolution) for generating images of similar resolution and better quality.

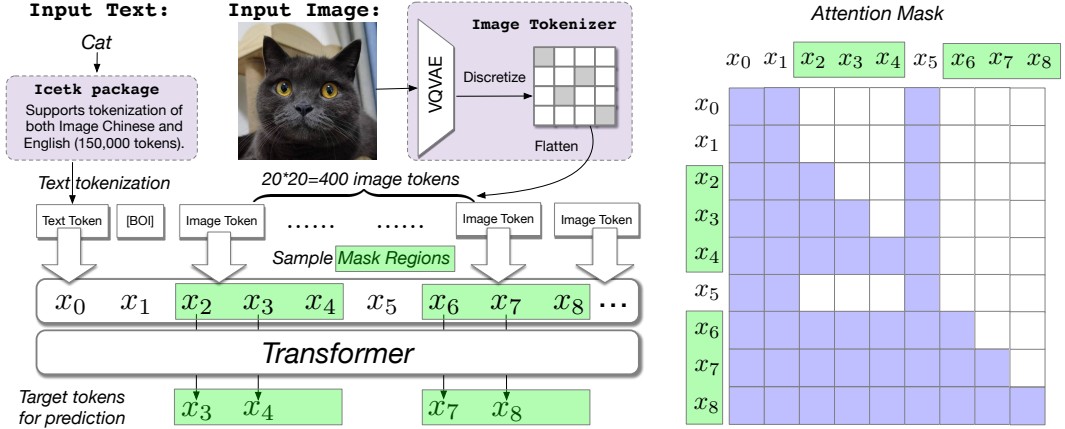

Figure 2: CogLM. (Left) The sequence consists of both text and image tokens. `[BOI]` (Begin-Of-Image) is the separator token. Mask regions are sampled according to different strategies. Only the second-to-last tokens in the mask regions are predicted to compute the loss. (Right) The mask regions are only implemented by changing the attention mask matrix, without any modification on the input tokens. In the attention mask matrix, rows and columns of all the masked tokens (the 2,3,4,6,7,8 rows and columns) can be extracted together to form a low-triangle attention mask matrix.

## 2 Related Work

**Text-to-image generation** for arbitrary inputs is a long-held dream for many cross-modal machine-learning researchers. Most early attempts to address this challenge were based on Generative Adversarial Nets [10]; these include AttnGAN [35], DM-GAN [40], DF-GAN [28], et al. Although they can perform vivid synthesis on domain-specific datasets, such as Caltech-UCSD Birds 200, general-domain datasets, such as MS COCO [17], present great challenges for these methods. DALL-E [26], CogView [3] and similar works [33, 8] leverage VQ-VAE [29] to compress an image to a sequence of discrete tokens and pretrain large transformers for autoregressive generation, greatly improving results in the general domain. LAFITE [39] learns to invert the pretrained CLIP [23] embeddings in the shared space of text and image for text-free training. Recently, many researchers have turned to diffusion models, largely due to the slow generation defect of autoregressive models. One example is Glide [19].

**Non-autoregressive generation** (NAR) is recently a popular topic in natural language generation—see Mask-Predict [9] and GLAT [21], which explores parallel decoding methods for autoregressive-like models. Generation speed was not an issue in the era when GANs dominated the image generation, but constitutes a considerable challenge for current autoregressive text-to-image models. M6-UFC [38] first introduces NAR methods into the VQ-VAE framework, and similar ideas are adopted by VQ-diffusion [11] and MaskGIT [1]. A possible drawback of pure NAR methods is that tokens sampled at the meantime might lead to global inconsistency in later steps during the generation of complex scenes. Our method introduces a hierarchical design to combine the consistency merit of autoregressive models and the speed advantage of NAR methods.

## 3 Method

### 3.1 The Cross-Modal General Language Model

While previous self-supervised pretext tasks often target at mask prediction in the computer vision [34, 12], our approach pursues a unification of autoregressive generation and bidirectional context-aware mask prediction.

In NLP, the General Language Model (GLM) [6] suggests changing the direct mask prediction into blockwise autoregressive generation. However, directly applying it to images would result in redundancy. For instance, the sizes of the masked image patches are fixed, thus we do not need the

capacity of filling blocks of indefinite length as in NLP. Moreover, GLM inserts a sentinel token for each mask region to predict its first token, which greatly increases the sequence length thus restricts the usage of 2D local attention.

Based on the analysis above, we present a simpler and more general language model for both text and image data—Cross-modal general Language Model (CogLM). As shown in Figure 2, CogLM takes as input a concatenation of text and images tokenized by *icetk* [1] (See § 3.2), whose dictionary contains 20,000 image tokens and 130,000 text (both Chinese and English) tokens. Formally, let $\mathbf{t} = [t_1, ..., t_M]$ be the text tokens and $\mathbf{im} = [im_1, ..., im_{N^2}]$ be the image tokens, where $M$ and $N^2$ are the lengths of text and image tokens respectively.

The crucial step in CogLM is to sample $k$ *mask regions* $R = \{[l_0, r_0], ..., [l_k, r_k]\}$ according to various strategies. In practice, the following two strategies are used:

- (Text-to-Image GPT) The input sequence is $\mathbf{x} = [\mathbf{t}\ \text{[BOI]}\ \mathbf{im}]$. We mask all the image tokens, which is similar to the pretraining task of CogView [3].
- (A Combination of Mask Prediction and Image Captioning) The input sequence is $\mathbf{x} = [im_0\ ...\ im_i\ ...\ im_j\ ...\ im_{N^2}\ \text{[BOE/C]}\ \mathbf{t}]$, where [BOE], [BOC] are separators meaning beginning-of-English and beginning-of-Chinese used for the corresponding language. we mask random patches and the text tokens. Ideally, the two tasks should be separated; but we combine them together for training efficiency.

Instead of replacing the tokens in the mask regions as [MASK], we make no change in the input but build an attention mask $A$ based on the mask regions. All tokens outside mask regions are seen as *context* and can be attended to by all other tokens. A token in mask regions can only be attended to by the tokens in mask regions and behind it. Specifically,

$$A[i,j] = \begin{cases} 1, & \text{if } \forall\, [l_u, r_u] \in R, j \notin [l_u, r_u], \\ 1, & \text{if } j \le i \text{ and } \exists\, u, v \text{ (indices)}, i \in [l_u, r_u] \in R, j \in [l_v, r_v] \in R, \\ 0, & \text{else.} \end{cases} \quad (1)$$

Figure 2 shows an example of the attention mask matrix of two mask regions.

In the mask regions, the model learns to predict the next token. The loss function can be written as follows:

$$L = \frac{-1}{\sum_u r_u - l_u} \sum_v \sum_{i=l_v}^{r_v - 1} \log p(x_{i+1} | x_{\le i}, x_{context}), \quad (2)$$

where the $x_{context}$ denotes the tokens outside the mask regions.

*A fox is siting on the books.*

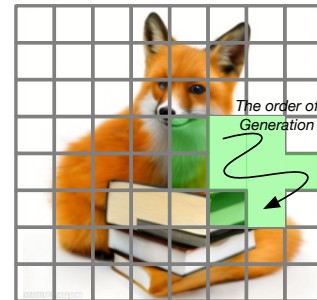

**Infilling.** Note that the first token in each mask region is not predicted during training. This feature seems to disable CogLM from image infilling or cloze filling in natural language, but this problem actually has a simple solution. During inference, we can move the last context token before each mask region into it, as illustrated in Figure 3. Although these moved tokens becomes *blind spots* for mask regions before them, they have few negative effects in practice. To further avoid this minor influence and fully maintain the context information, we deal with each mask region individually. For each region, we move only the last context token before *this* region, and keep all the known tokens outside the mask regions. Thus, we cannot use the cached hidden states from the previous region, slightly slowing down the multi-region infilling. See Appendix A for samples.

Figure 3: Image Infilling of CogLM. Tokens (viewed as patches) in light green mean mask regions.

**Advantages over GPT [22], GLM [6] and MAE [12]. (GPT)** The main advantage over GPT is that the modeling of bidirectional contexts is considered in CogLM,

[1] http://github.com/THUDM/icetk

which will benefit many tasks relying on global information, e.g. super-resolution in the next section and image classification. The importance of bidirectional context has been verified in the comparison of BERT [2] and GPT on GLUE [31]. **(GLM)** The main advantage over GLM is simplicity. To unify the generation and bidirectional understanding, GLM needs to define many new special tokens and a new type of position embedding, insert a sentinel for each mask region and change the order of input tokens. It destroys the spatial relevance in the image data and excludes the possibility of using 2D local attention or convolution. **(MAE)** MAE is designed for self-supervised learning on pure image data and is not ready for generation. Even without text, CogLM is more parameter-efficient, because MAE is an encoder-decoder structure. A considerable part of parameters in encoders and decoders are learned for the same function, e.g. extracting basic features from inputs.

## 3.2 Pretraining

As we have introduced CogLM as a general pretraining framework, in this section, we will describe the details and hyperparameters of our pretrained CogLM.

**Tokenization.** We have developed a unified tokenizer *icetk* of **I**mage, **C**hinese and **E**nglish. As shown in DebertaV2 [13], a large vocabulary (128,000 tokens) offers many benefits. For text, we extract a bilingual vocabulary of 130,000 tokens in icetk and explicitly classify them as Chinese, English, Common or Rare Symbols, so that we can specify the generated language via a sampling mask. The image tokenizer is a 20,000-token first-stage VQ-VAE [29], largely following the tokenizer in CogView [3]. Inspired by Esser et al. [7], a term of perceptual loss [37] is added to the reconstruction loss, significantly improving reconstruction performance. (See Appendix for details.)

**Transformer.** The backbone of our pretrained CogLM is a Transformer with Sandwich Layer-Norm [3]. The model has 6 billion parameters (48 layers, hidden size 3072, 48 attention heads), trained for 300,000 iterations in FP16 with batch size 4,096. The sequence length is 512, consisting of 400 image tokens, 1 separator and up to 111 text tokens.

**Masking Strategy.** We randomly select a sampling strategy for each training sample. For the mask prediction strategy, the analysis from SimMIM [34] exhibits the great importance of mask percentage and patch distribution. We follow their results to sample $4 \times 4$ token patches at random until 75% of the tokens are in the mask regions. For bilingual samples, we randomly choose one of the languages during training.

## 3.3 Hierarchical Generation

Although the pretrained CogLM can generate images from text, the resolution is only $20 \times 20$ tokens ($160 \times 160$ pixels). The short sequence is intentional, for fast generation. The versatility of CogLM allows us to fine-tune it into super-resolution models. The whole hierarchical pipeline makes up our CogView2 system.

**Direct super-resolution.** In this step, we want a model to map a generated low-resolution image token sequence $\mathbf{im}^0 \in [0, 20000)^{20 \times 20}$ to a higher-resolution sequence $\mathbf{im}^1 \in [0, 20000)^{60 \times 60}$. We fine-tune the pretrained CogLM into an encoder-decoder architecture. The input of the encoder is the $20 \times 20$ sequence of generated image tokens, and the input of the decoder is just a $60 \times 60$ sequence of [MASK]. We do not follow the original transformer [30] to add a cross-attention layer, instead we make the tokens in the decoder attend both local tokens in decoder and encoder. This cross-resolution local attention is implemented via a customized CUDA kernel introduced in section 4.2. Both encoder and decoder are initialized using the pretrained CogLM. In practice, we find it enough to only fine-tune the weights of the attention layers in the decoder, so that we can fix and share the other parameters between the encoder and decoder to reduce the memory consumption.

Although direct mapping is a traditional practice for super-resolution—e.g. SRCNN [4]—it is hardly qualified as generation; it focuses more on texture transformation. The loss function of direct mapping is token-based or pixel-based (MAE), meaning that it predicts or maximizes the marginal distribution $p(im_i^1 | \mathbf{im}^0)$ for each token $i$ instead of $p(\mathbf{im}^1 | \mathbf{im}^0)$. As we use cross-entropy loss and a multinomial sampling during generation, we get

$$\mathbf{im}^1 = [im_1^1, ..., im_{60 \times 60}^1], im_i^1 \sim p_\theta(im_i^1 | \mathbf{im}^0), im_i^1 \text{ and } im_j^1 \text{ are independent if } i \neq j. \quad (3)$$

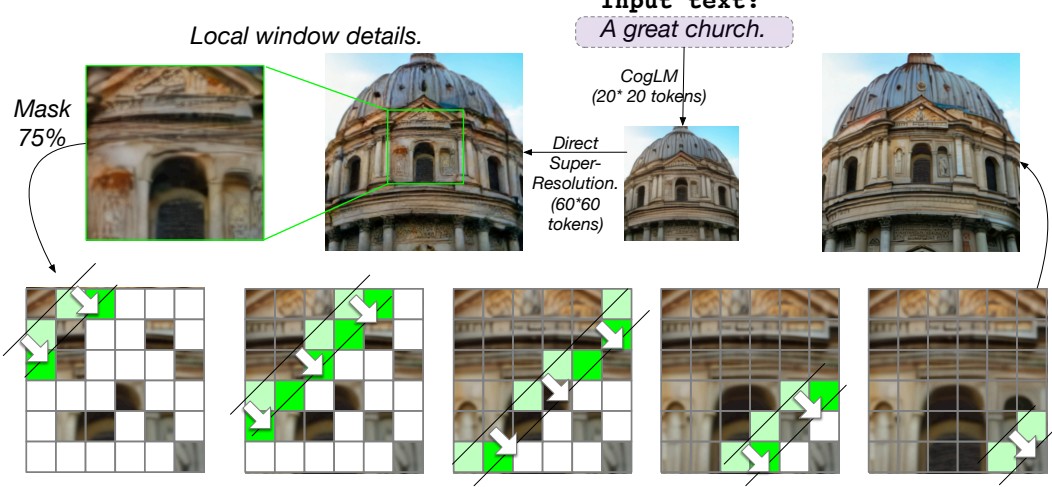

Figure 4: Super-resolution modules. Low-resolution images are mapped into high-resolution images via the direct super-resolution module. In each snapshot during the iterative super-resolution, all tokens of the same color are generated at the same time. All the local windows work in parallel.

Therefore, we need to refine $\mathbf{im}^1$ using another module.

**Iterative super-resolution.** In this step, we aim to refine the initial high-resolution sequence $\mathbf{im}^1$ into a better one $\mathbf{im}^2$. The working principle of the refinement is to break the independence of the generated tokens, while keeping the parallelism. Thus, we propose a local parallel autoregressive (LoPAR) approach.

The motivation of LoPAR is that the hierarchical process frees us from global dependence. As long as we maintain 25% – a ratio from MAE [12] – random tokens as context, it is sufficient to recover the global scene of the image. If the re-generated tokens are coherent locally with 25% kept tokens, global coherence is also guaranteed. We mask 75% of the tokens of $\mathbf{im}^1$ and assume that there is a local window size $\sigma$,

$$p(\mathbf{im}_i^2|\mathbf{im}^1) = p(\mathbf{im}_i^2|\{\mathbf{im}_j^1 \mid \text{dist}(i,j) < \sigma \text{ and } j \text{ is not masked.}\}), \qquad (4)$$

$$p(\mathbf{im}_i^2|\mathbf{im}^1, \mathbf{im}_j^2) = p(\mathbf{im}_i^2|\mathbf{im}^1) \text{ if } \text{dist}(i,j) > \sigma, \qquad (5)$$

so that local attention is sufficient and tokens from different local windows can be generated in parallel. To further increase the parallelism, we find the local inconsistency usually occurs when directly adjacent (vertically or horizontally) tokens are generated at the same time. We factorize the generation process into different iterations diagonally as in Figure 4 and below:

$$p(\mathbf{im}^2|\mathbf{im}^1) = \prod_{k=0}^{2\sigma-1} \left( \prod_{i}^{\text{row}(i)+\text{col}(i)=k} p(\mathbf{im}_i^2|\mathbf{im}^1, \{\mathbf{im}_j^2 \mid \text{row}(j) + \text{col}(j) < k\}) \right), \qquad (6)$$

where $\text{row}(i) = \lfloor \frac{i-1}{60} \rfloor \mod \sigma$ and $\text{col}(i) = (i-1) \mod \sigma$ are the indices of row and column in the local window.

To implement the iterative super-resolution module, we fine-tune the pretrained CogLM for 20,000 iterations into a BERT-style masked prediction model on $60 \times 60$-token sequences with local attention. The mask ratio is sampled from $\{0.2, 0.4, 0.6, 0.8, 0.9\}$ for each sample. During inference, we set the local window size to $\sigma = 6$ and compress the iterative process from $2\sigma - 1$ to 6 iterations by arranging the unmasked tokens and merging the first and final iterations[2].

---

[2]Implemented by a manually designed $6 \times 6$ matrix. Details are included in our released codes.

# 4 Plug-in Improved Techniques for Transformers

## 4.1 Cluster Sampling

In autoregressive generation, the sampling strategy over the predicted distribution of the tokens is crucial. Top-k or top-p (nucleus) sampling [14] are the most common strategies, but suffer from an *incomplete truncation* problem.

The vocabulary of the image tokens is learned by VQVAE [29], where the embeddings of some tokens are very similar. To represent the frequent patterns at a finer granularity, we use a large vocabulary of 20,000 tokens, three times larger than that of the previous works [26, 3], further exacerbating the situation. For instance, there are about 42 tokens basically "white" in *icetk*, which show subtle differences only when connected to some other tokens. Although the sum of the probabilities of these "white" tokens might be large enough, most of them could be filtered by top-k sampling. Figure 5 illustrates the problem.

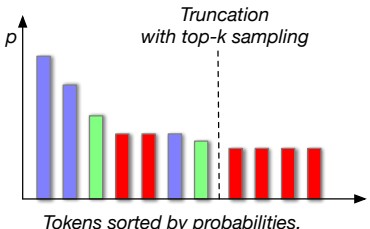

Tokens sorted by probabilities.

Figure 5: (Best viewed in color.) Incomplete truncation. The same color indicates very similar embeddings of the tokens. The hard truncation of top-k sampling twists the proportion between blue, green and red tokens.

To solve the incomplete sampling problem, we propose cluster sampling. We group the 20,000 tokens into 500 clusters via K-means [18] based on their vectors in VQVAE. During sampling, we first sample a cluster using top-k sampling based on the sum of probabilities of tokens in the clusters, and then sample in the cluster. All the tokens within a cluster are treated as a whole and will be filtered or kept together, alleviating the incomplete truncation problem.

## 4.2 Local Attention

Locality is one of the most important properties of image data. Local operations, e.g. convolution, dominated the visual computing before ViTs [5]. Even attention in the ViTs mainly deals with the interactions between local tokens [24]. We find it possible to fine-tune the pretrained CogLM using local attention and textual attention, which is generally compatible with the global attention weights from pretraining. However, 2D local attention cannot be implemented efficiently using high-level framework, e.g. Pytorch [20]. We develop a customized CUDA kernel to support both 2D local attention, 2D autoregressive local attention and cross-resolution local attention. In the CUDA kernel implementation, we can save half of the computation in the matrix multiplication and do not need a causal attention mask for the autoregressive attention. In the super-resolution modules, we use local attention with the receptive field (RF) of $9 \times 9$. Figure 6 show the benchmark for a single-head attention with hidden size 64 on a A100 GPU. The advantage of our method will be more obvious in autoregressive scenarios, which is up to $40\times$ faster and consumes $1\%$ memory than global attention on 4,096 sequences.

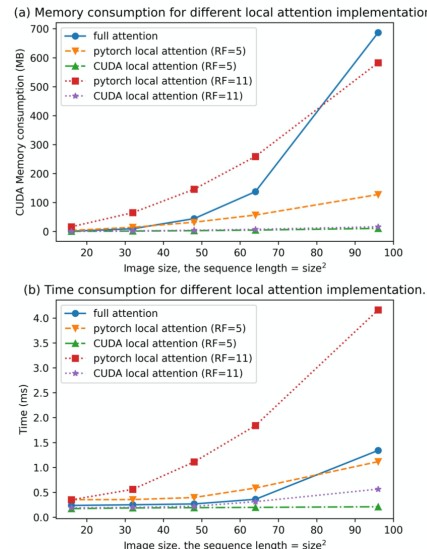

Figure 6: Comparison between CUDA kernel-based local attention, full attention, and Pytorch implementation based on the `unfold` (im2col [15]) operation. The hidden size in the benchmark is 64.

## 4.3 Upweighting Textual Attention

Most text-image pairs are weakly relevant in the large training data of CogLM. Even the model perfectly fits the data, it should have a considerable probability to generate irrelevant images. To strengthen the relevance, we leverage the explainability of the attention operation. We add a constant $c$ to the attention scores from any token to the text tokens: (The attention mask is omitted for simplicity)

$$\text{Attention}(Q, K, V, A) = \text{softmax}(\frac{Q^T K}{\sqrt{d}} + [\underbrace{c \dots c}_{text\ part} \quad \underbrace{0 \dots 0}_{image\ part}])V. \tag{7}$$

This technique costs ignorable time consumption but largely improves the textual relevance of the generated images. In practice, $c < 3$ will not influence the quality of the images.

## 5 Experiments

### 5.1 Dataset

Our dataset for pretraining contains about 30 million text-image pairs, mostly overlapping with that of CogView [3]. We filter about 5 million text-image pairs from the CogView dataset with some keywords, e.g. "abstract" and "texture", because they are mostly background images used for design. These images consist of repeating patterns and contribute little to text-to-image generation. We then replenish the dataset with 5 million tag-image pairs. About half the text is translated from English, and both Chinese and English text are kept to train our bilingual CogLM. Only the images whose resolution is at least $480 \times 480$ are used to train the super-resolution modules.

### 5.2 Machine Evaluation

To compare with previous and concurrent works, we follow the most popular benchmark originated from DALL-E [26], Fréchet Inception Distances and Inception Scores evaluated on MS-COCO [17]. 30,000 captions from the validation set are sampled to evaluate the FID. Since each image in COCO has up to 5 different captions, we carefully select the sampled captions to describe different images. We generate 16 samples for each caption (translated into Chinese), and select the best one with the lowest caption perplexity (the Caption Score in [3]). Note that FID is not the perfect metric to evaluate CogView2 because (1) the advantage of CogView2 is to generate high-resolution images, but we need to resize the images back to $256 \times 256$ for meaningful comparison. (2) There are mistakes when translating English captions into Chinese. (3) Our training data contain many single-object images, which are quite different from those in the distribution of COCO (common objects *in context*).

Table 1: Machine Evaluation Results on MS-COCO. (Downsampling CogView2 images to $256 \times 256$.) "*" means fine-tuning on MS-COCO. "– technique" is the ablation study without this technique. CogView2 achieves **the best blurred FIDs** over all comparable methods.

| Model | FID-0 | FID-1 | FID-2 | FID-4 | FID-8 | IS |
|-------|-------|-------|-------|-------|-------|-----|
| AttnGAN* [35] | 35.2 | 44.0 | 72.0 | 108.0 | 100.0 | 23.3 |
| DM-GAN* [40] | 26.0 | 39.0 | 73.0 | 119.0 | 112.3 | **32.2** |
| DF-GAN* [28] | 26.0 | 33.8 | 55.9 | 91.0 | 97.0 | 18.7 |
| DALL-E [26] | 27.5 | 28.0 | 45.5 | 83.5 | 85.0 | 17.9 |
| CogView [3] | 27.1 | 19.4 | 13.9 | 19.4 | 23.6 | 18.2 |
| XMC-GAN* [36] | **9.3** | - | - | - | - | 30.5 |
| NÜWA* [33] | 12.9 | 13.8 | 15.7 | 19.3 | 24 | 27.2 |
| LAFITE [39] | 26.9 | 23.0 | 18.7 | 15.7 | 14.8 | 26.0 |
| VQ-diffusion-F* [11] | 13.86 | - | - | - | - | - |
| Make-A-Scene* [8] | **7.55** | - | - | - | - | - |
| DALL-E-2 [27] | 10.9 | - | - | - | - | - |
| CogView2 | 24.0 | 19.7 | 16.8 | 17.2 | 17.2 | 22.4 |
| – clustering sampling | 36.4 | 32.4 | 28.9 | 28.5 | 30.4 | 18.8 |
| – attention upweighting | 24.6 | 20.4 | 17.5 | 17.9 | 18.9 | 21.1 |
| CogView2* | 17.5 | **13.4** | **10.9** | **10.6** | **10.4** | 25.2 |

The results of machine evaluation are demonstrated in Table 1. We find that fine-tuning CogLM on the MS-COCO dataset will largely improve the FID. During our fine-tuning, FID diminishes from 24.0 (0 iteration)$\rightarrow$ 19.2 (2,500 iterations) $\rightarrow$ 17.5 (7,500 iterations). However, we find that the quality (human evaluation) of generation deteriorates. Though the style is similar to COCO, the

generation is not as accurate as for the non-fine-tuned version, which also corresponds to the scores in human evaluation in Figure 7.

## 5.3 Human Evaluation

As the most persuasive metric, we conduct a large-scale human evaluation following the setting in CogView [3] (See Appendix for details). The experiments include a total of 4,600 groups of comparison on COCO captions between some public available text-to-image works, including DF-GAN [28], LAFITE [39], CogView [3], CogView2 (including its finetuned version on COCO) and the recovered ground truth after VQVAE. Note that the VQVAE in CogView2 is much better than that in CogView, which makes the recovered ground truth a stronger upper bound. The results are demonstrated in Figure 7. An intriguing finding is that the finetuned CogView2, although with much better FID, performs worse than the original model. We guess that the model might fit the style of complex scenes in COCO, but the generated samples with isolated subjects could be preferred by the annotators.

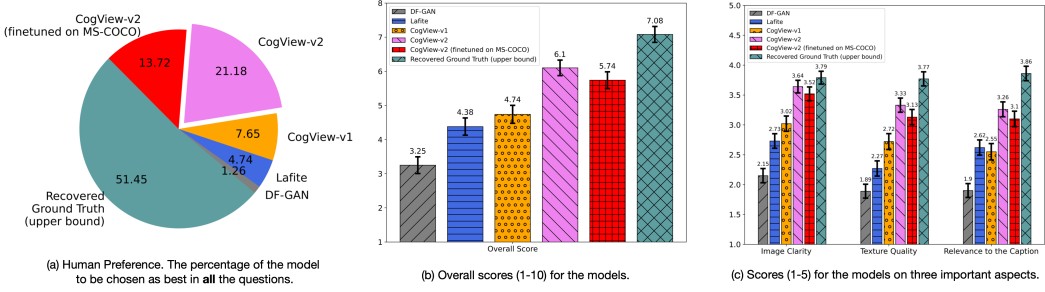

Figure 7: The results of human evaluation. CogView2 performs the best in all the aspects.

## 5.4 Analysis of the Speed and FLOPs of LoPAR

As discussed in § 1, our motivation is to increase the degree of parallelism for inference acceleration, even with more FLOPs. Autoregressive generation with cached hidden states have the same FLOPs with a teacher-forcing forward step, but is much slower (858ms vs 225.9s in CogView2 scale). For LoPAR, it is exactly $N$ ($N = 6$ in our setting) times and FLOPs of forward steps. We compare the inference speed of super-resolution stage with different strategies in Table 2.

Table 2: The wall-clock time and FLOPs for a 4,096 sequence on an A100-40GB GPU with different AR-related methods. The model configs are the same as the pretrained CogLM 6B.

|  | FLOPs | Time | Memory (inference) |
| --- | --- | --- | --- |
| Forward (also teacher forcing training) | $1.17 * 10^{14}$ | 858 ms | 5,041MB |
| Autoregressive generation (no cache) | $4.81 * 10^{17}(4095\times)$ | about 1h | 5,041MB |
| Autoregressive generation (cached) | $1.17 * 10^{14}(1\times)$ | 225.9s | 4,865MB |
| LoPAR | $7.02 * 10^{14}(6\times)$ | 4.89s | 5,041MB |
| LoPAR + local attention | $5.82 * 10^{14}$ | 3.41s | 352MB |

## 6 Discussion

**Autoregressive or Diffusion?** Although GPTs achieved great success in text generation, diffusion models are becoming increasingly popular in image generation. Here we compare diffusion models with autoregressive models from the aspect of speed, the largest disadvantage of the autoregressive models discussed in the section 1. With the same architecture, diffusion models require more FLOPs but have a high degree of parallelism. They can also make a trade-off between the quality and time consumption by manually scheduling the stride of sampling. For example, Glide [19] samples 250 diffusion steps for evaluation, and 27 steps for interactive sampling, to reduce the latency to 15s.

Autoregressive models must generate the image token-by-token, but our LoPAR can upsample the image with a high parallelism degree, so that (potentially) we can reduce the time cost by introducing more hierarchies to design models much faster than diffusion models.

**Comparison between DALL-E-2 and CogView2.** DALL-E-2 [27] is a recently released concurrent work for text-to-image generation on $1024 \times 1024$ resolution. Its probabilistic model and architecture are quite different from those in CogView2. But both models share the same spirit – hierarchical generation. The difference is that DALL-E-2 adopts an additional third-level super-resolution and a generation prior, which help result in potential quality gain, but also lead to expensive resource-consuming. CogView2 is able to synthesize similar scenes according to the limited demos of DALL-E-2, e.g. "lion teacher" (Figure 1) vs. "panda scientist" (DALL-E-2), considering CogView-2 is trained using only 5% of the total data (650M text-image pairs) by DALL-E-2. For future, CogView2 can also adopt the third-level super-resolution and the prior, though it is engineering mostly.

## 7 Conclusion

The breakthrough in the text-to-image domain is made by autoregressive models. However, the slow generation and high complexity hinder researchers attempts to improve the quality in this direction. In this paper, we put forward an approach based on hierarchical transformers to help autoregressive models remedy these disadvantages, and bridge the gap between text-to-image pretraining and recent visual representation learning methods.

**Broader Impact.** The advancement of text-to-image generation, especially text-guided image editing, will ease the creative efforts of artists and designers, while also causing a risk of misinformation, leading to permanent damages to the reliability of web photos. However, it is possible to train a classifier to distinguish the real and CogView2-generated images according to the texture features.

## Acknowledgments and Disclosure of Funding

We would like to thank Zhao Xue and Sha Yuan for the help on collecting the dataset, Hanxiao Qu for maintaining the machines, and Yue Cao and Chang Zhou for their useful discussion, Zhendong Zhang for releasing an initial version of CUDA local attention.

Funding in direct support of this work: GPU hours donated by BAAI, NSFC for Distinguished Young Scholar (61825602).

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
