# A High-resolution Text-guided Infilling

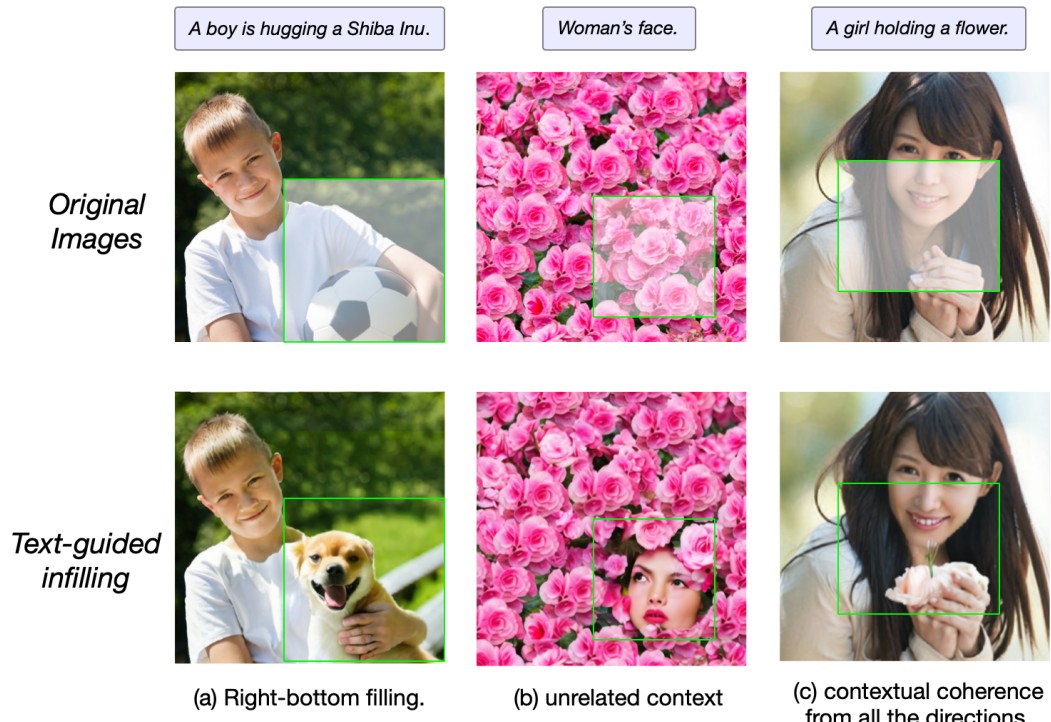

Figure 8: Examples of text-guided infilling. (a) The traditional right-bottom filling. This is also supported by ordinary GPTs, but GPT-based models cannot see the right-bottom context as in (c) during generation. (b) Generating objects in an unrelated context. (c) CogView2 can keep the coherence from all the directions, e.g. the face on the top and the fingers at the bottom.

Although CogLM in nature supports text-guided infilling as illustrated in Figure 3, we find that for small patches, the model tends to only consider the coherence of context and ignore the text, because the granularity of the CogLM is only $20 \times 20$. Our solution is to magnify the region to an appropriate size and then run the infilling and super-resolution. Specially, the process can be divided into the following steps:

1. Calculate the square bounding box of the given mask region.

2. Scale the size of the bounding box by 1.4 (as a default hyperparameter).

3. If the box already covers the whole image, run the normal completion by CogLM and super-resolution; else magnify the patch in the boxed region to $480 \times 480$, run the normal completion by CogLM and super-resolution only for this region, and scale the result back to the original size.

4. Replace the masked region to the generated results.

5. Tokenize the resulted image and de-tokenize it immediately to keep the coherence at the edge of the masked region.

We show a few examples of text-guided infilling in Figure 8. The examples show the potential of CogView2 on the applications like portrait editing.

# B Details of the Tokenizers

Here we introduce more details about our unified tokenizer, icetk.

## B.1 Text Tokenizer

The text tokenizer is trained based on the sentencepiece [3] (unigram algorithm [16]) on a mixed corpus of both English and Chinese. The corpus consists of 25GB plain text – half English, half Chinese. We divide the extracted tokens into four categories: common, English, Chinese, and special, by carefully selection according their unicode encoding, and assign them with consecutive ID numbers.

The dictionary size of icetk is 150,000. The first 20,000 tokens are image tokens (discussed below). There are 100 common tokens, including punctuations, numbers and self-defined tokens, e.g. <unk>. The No. 20,100 to No. 83,822 tokens are English tokens and the No. 83,823 to No. 145,653 tokens are Chinese tokens. The rest are special tokens, e.g. $\alpha$. We can disable the generation of part of tokens via a sampling mask, e.g. only generating English tokens for image captioning.

## B.2 Image Tokenizer

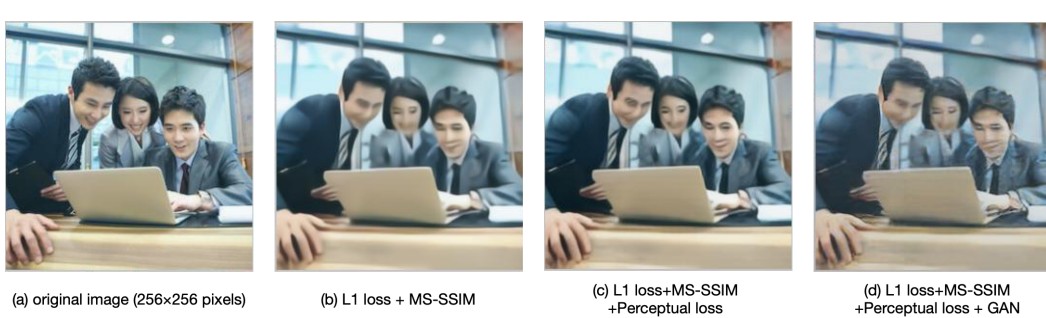

| (a) original image (256×256 pixels) | (b) L1 loss + MS-SSIM | (c) L1 loss+MS-SSIM +Perceptual loss | (d) L1 loss+MS-SSIM +Perceptual loss + GAN |

Figure 9: Comparison between the reconstruction ($8\times$ compression) with image tokenizers trained with different losses. (d) can be seen as a reproduction of VQGAN [7]. We can find that the perceptual loss (c) improve the quality over (b). VQGAN (d) reconstructs images with more details, but sometimes not very accurate, e.g. the mouth in the example.

The image tokenizer is a multi-compression-rate VQVAE [4]. The main differences between the image tokenizer of CogView2 and that of CogView are mainly the *perceptual loss* and the *multi-compression-rate* design. The perceptual loss [37] is proposed to measure the matching degree of the human

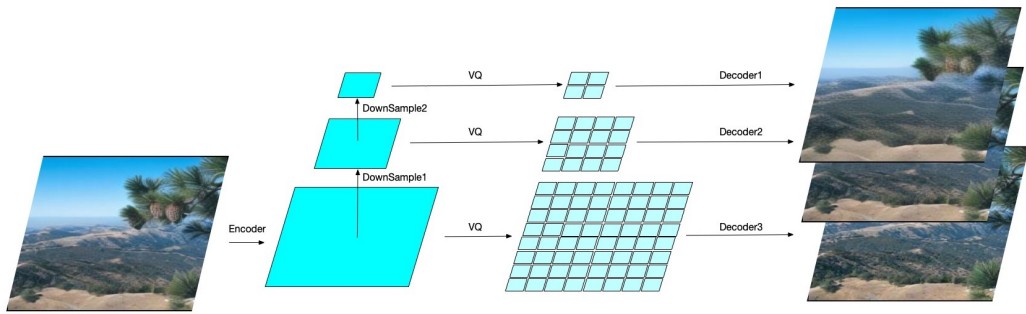

Figure 10: The illustration of our multi-compression-rate design. We find the pooling operation performs best as the DownSample layers.

perception and the texture of a given image. VQGAN [7] uses it as one of the loss term for GAN training. We surprisingly find that the perceptual loss might account for the majority of the texture improvement of VQGAN over the VQVAE. However, VQGAN, although proved to achieve better FID, sometimes produces bad cases on important elements of the images, e.g. the face. Figure 9

---

[3]http://github.com/google/sentencepiece

[4]The term "VQVAE" originally [29] refers to the whole process of compression of discrete latent variables (the first stage) and the autoregressive modeling of the prior (the second stage). In this paper, we refer the term "VQVAE" only to the first stage for simplicity.

shows an example for comparison between the methods. Therefore, we finally determine to use the $L_1$, MS-SSIM [32] and perceptual loss to train a VQVAE as the image tokenizer as follows,

$$L_{rec}(x) = |x - \hat{x}| + \text{MS-SSIM}(x, \hat{x}) + \text{Perceptual-Loss}(x, \hat{x}),$$

where $\hat{x}$ is the recovered output after the VQVAE.

The multi-compression-rate design means that our image tokenizer can compress the image with different compression rates, specifically $4^2\times, 8^2\times$ or $16^2\times$. Although all of our experiments in CogView2 only use the $8^2\times$ compression rate, i.e., $160^2 \rightarrow 20^2$ and $480^2 \rightarrow 60^2$. We implement it by training three image tokenizer with a shared dictionary and low-level-layer parameters, which is illustrated in Figure 10. The architecture basically follows the released VQGAN [7] with self-attention layers. We train the model on ImageNet to convergence.

## C   Details about Human Evaluation

We follow the setting of CogView [3] to conduct human evaluation. 50 captions from MS COCO are randomly selected. The evaluators need to score each image (from 1 to 5) from three aspects: the image clarity, the texture quality and the relevance to the caption, and give an overall score from 1 to 10. The evaluators are also asked to select the best one from the images.

We recruited 100 evaluators, and 92 of them finally finished the tasks. Each evaluator will score 300 images and get paid with 75 yuan. The screenshot for the evaluation website are shown in Figure 11. We also find a very small part of participants did not carefully score the images, but randomly select it to quickly finish the task. This may account for that even DF-GAN wins in 1.26% questions.

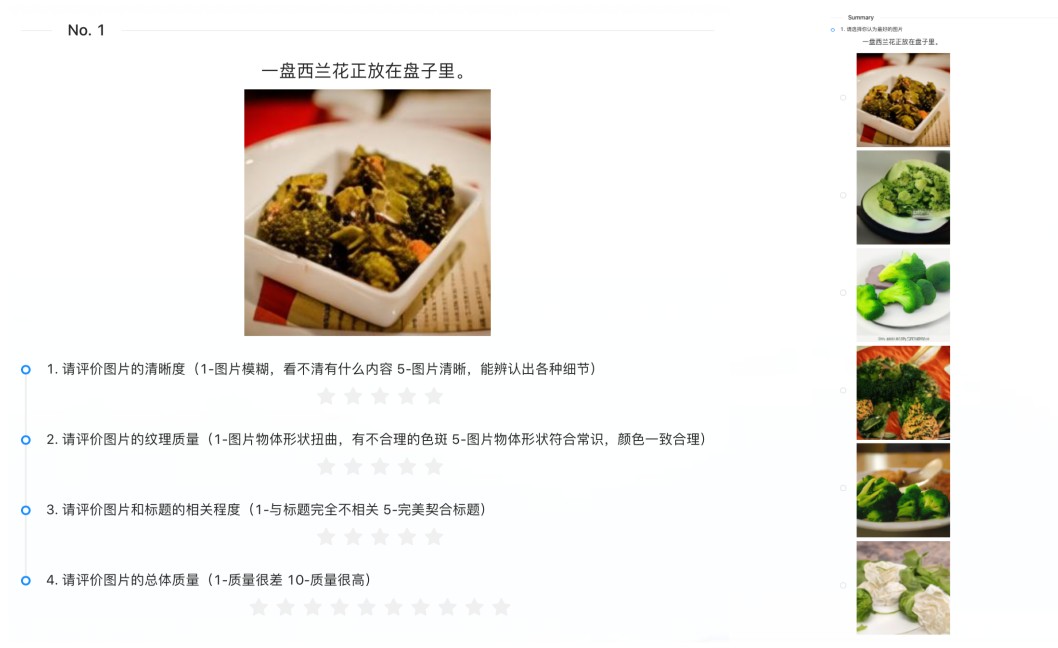

(a) Rate a single sample.      (b) Select the best one for a given caption.

Figure 11: Screenshots of human evaluation. For each caption, the evaluator will rate each sample from different methods as in (a), and finally choose the best one (b).