# OpenReview forum: "CogView2: Faster and Better Text-to-Image Generation via Hierarchical Transformers"
_NeurIPS.cc/2022/Conference — NeurIPS 2022 Accept_

### Official Review · Reviewer_GY7Q · 2022-07-10

**Rating:** 6
**Confidence:** 4
**Soundness:** 2 fair
**Presentation:** 3 good
**Contribution:** 2 fair

**Summary:**

This paper proposes a faster and better text-to-image generation model called CogView2. Compared to the CogView baseline, main contributions are: 1) hierarchical generation that upsamples the original low-resolution image tokens and then refine them, 2) customized CUDA kernel that speeds up training, 3) a special attention masking strategy used in CogLM during training. Experiments clearly demonstrate that CogView2 generates better images than CogView both in terms of FID scores and also in terms of human evaluation.


**Questions:**

(1) It will be great if authors can provide more visualizations, especially those with higher resolution. I can not judge the quality of generated images from just a few hand-picked ones.

(2) What is FID-k in Table 1? I assume it means radius of the Gaussian Filter but couldn’t find any explanation in the text. If my guess is correct then shouldn’t FID–0 be the most important number? Does this mean that CogView2 is worse than multiple baseline methods in terms of image quality?

(3) As stated before, I have concerns about the masking strategy. I do not see a motivation for joint learning of autoregressive generation and bidirectional mask prediction. Authors are encouraged to provide more ablation studies to support the design of CogLM and explain why it is better than the training in CogView.

(4) For human quality evaluation, I wonder why authors do not include DALL-E as part of the test? That should be a key baseline.



**Limitations:**

Limitations and potential negative societal impact are adequately addressed by the authors.


**Strengths And Weaknesses:**

Strength:

The paper is mostly easy to follow. Authors’ proposed idea of using super-resolution and refinement modules to hierarchically generate higher resolution images is intuitive and works well in practice. Designing these two modules is clearly non-trivial work. Authors further demonstrate that clustering sampling is better than simple top-k. I also appreciate that they are willing to write a customized cuda kernel to speed up training. The speed-up seems to be significant in the autoregressive case.

Weakness:

Model improvement is limited compared to CogView. Both use Transformer to jointly learn the likelihood of text tokens and image tokens. The main difference is the new masking strategy in CogLM where tokens inside the mask region are trained to predict the next token based on past masked-tokens and non-masked context tokens. Authors stated that this approach unifies autoregressive generation and bidirectional prediction, but I find this design lacking justification. Experiment section also fails to provide any ablation study to support the choice of this particular masking strategy.

Even with the new hierarchical design, CogView2 still generates image resolution lower than other works such as DALL-E-2 [1] and Imagen [2]. Stacking another direct/iterative super-resolution module is straightforward and should solve the issue. Resource limitation is indeed a problem but still I need to point out the lower resolution as part of the weakness.

Only a few hand-picked visualizations are provided in the main paper and in the supplement. This makes it very hard to qualitatively judge the performance.

[1] Ramesh, Aditya, et al. "Hierarchical text-conditional image generation with clip latents." arXiv preprint (2022).

[2] Saharia, Chitwan, et al. "Photorealistic Text-to-Image Diffusion Models with Deep Language Understanding." arXiv preprint (2022).

---

> ### Author Response · Authors · 2022-08-02
> **Response**
>
>
> Thank you very much for your review. We will explain your concerns point by point.
>
> > The main difference is the new masking strategy in CogLM where tokens inside the mask region are trained to predict the next token based on past masked-tokens and non-masked context tokens.
>
> Please also consider the LoPAR upsampling method, which generates samples much faster than the pure auto-regressive way for high-resolution images. Our local attention kernel, attention upweighting, clustering sampling can also benefit the other auto-regressive generative models.
>
> >  It will be great if authors can provide more visualizations, especially those with higher resolution. I can not judge the quality of generated images from just a few hand-picked ones.
>
> Of course, we can. There are some unfiltered batches of images in different categories.
>
> - people: https://imgur.com/uM3LWLM
> - scenes: https://imgur.com/cyznGD4
> - animals: https://imgur.com/TIZT97C
> - objects: https://imgur.com/bH9FJ9G
> - scientific: https://imgur.com/pQSxMcd
>
> > What is FID-k in Table 1? I assume it means radius of the Gaussian Filter but couldn’t find any explanation in the text. If my guess is correct then shouldn’t FID–0 be the most important number? Does this mean that CogView2 is worse than multiple baseline methods in terms of image quality?
>
> The FID-k is the metric proposed in the DALL-E paper[1]. It blurs the details and compare the main contents. FID-0 cares a lot about details. For example, DALL-E has worse FID-0 (and better FID-k) than DM-GAN and DF-GAN, but it is still seen as a great advancement and get higher human evaluation scores. So FID–0 is not always the most important number. A recent work (https://www.cs.cmu.edu/~clean-fid/) also reveals that even the jpeg quality has a great influence on FID-0.
> In our experiments, we care more about human evaluation performance, where CogView2 outperforms CogView, LAFITE et al. by a large margin.
>
> [1] Ramesh, Aditya, et al. "Zero-shot text-to-image generation." 2021.
>
> > As stated before, I have concerns about the masking strategy. I do not see a motivation for joint learning of autoregressive generation and bidirectional mask prediction. Authors are encouraged to provide more ablation studies to support the design of CogLM and explain why it is better than the training in CogView.
>
> As discussed in Line 31-39, we want to overcome the defect of unidirectionality of usual auto-regressive models. CogLM naturally supports text-guided infilling tasks, as showed in Appendix B, and make the finetuning for itersr very easy, because the itersr task is very similar to the mask prediction in the CogLM pretraining.
>
> To train a versatile transformer in a simple way itself is also a popular topic. For example, the most recent work (28, July) of OpenAI proposed an extremely similar model as CogLM, called FIM[1] to fulfill infilling jobs, and verify its performance for left-to-right generation.
>
> [1] Efficient Training of Language Models to Fill in the Middle. https://arxiv.org/pdf/2207.14255.pdf
>
> > For human quality evaluation, I wonder why authors do not include DALL-E as part of the test? That should be a key baseline.
>
> Because DALL-E is not open-source. Many text-to-image models are not open-source, so that we cannot include them in the evaluation. This also highlights the value of open-sourcing of CogView2.
>
> If our answer above ease your concerns, could you increase your rating a bit? Please tell us if you have further concerns.

---

> ### Comment · Reviewer_GY7Q · 2022-08-08
> **Post-rebuttal response**
>
> I appreciate authors for providing the extra visualizations. Most of my questions are resolved, but still I am not very convinced but the masking strategy. I raise the score to weak accept based on overall contributions.

---

### Official Review · Reviewer_swZ7 · 2022-07-10

**Rating:** 8
**Confidence:** 3
**Soundness:** 4 excellent
**Presentation:** 4 excellent
**Contribution:** 3 good

**Summary:**

This paper proposes a pretraining method, a Cross-Modal General Language Model (CogLM), that masks both image and text tokens in input and learns to predict them in an autoregressive manner, while handling bidirectional context. By fine-tuning a pretrained transformer with this approach, the authors construct a hierarchical model, CogView2, which first maps a generated image into a larger image (direct super-resolution) and refines local patches (local parallel autoregressive), thus improving resolution as well as inference speed.

The experiments in the paper suggest that CogView2 performs comparable to other models despite its smaller model size and training data size. Meanwhile, the approach results in a considerable reduction in model run times (e.g. 10x faster than its predecessor, CogView).

Training and evaluation in this paper considers text in both English and Chinese.


**Questions:**

Questions
* What were the main challenges/blockers for directly comparing different models’ inference time in an end-to-end fashion?

Suggestions
* Discussion on failure modes: Even from the cherry picked examples in Figure 1, there are multiple failure modes observed (e.g. different numbers or lengths of fingers). What are the main types of failure modes the authors or human annotators observed? Any difference when it’s in English vs. Chinese?

Nitpicking
* In Figure 2: “Supports tokenization of both Image Chinese and English” → “Supports tokenization of both images and texts in Chinese and English”
* In Section 2: “DF-GAN, et al.” → “and DF-GAN.”
* In Section 2 and throughout the paper: use either “VQ-VAE” or “VQVAE” to be consistent
* In Section 3.1: consider moving the paragraph “In NLP, the General Language Model [...]” to Related Work
* In Section 3.1: $l$ and $r$ are not defined
* In Section 3.1: “where [BOE], [BOC] are separators meaning beginning-of-English and beginning-of-Chinese” → “where [BOE] and [BOC] are separators to indicate the beginning of English text and that of Chinese text”
* In Section 3.1: “Ideally, the two tasks should be separated” is not justified
* In Section 3.2: “Image, Chinese and English” doesn’t really type check; maybe it should be “Image and Text in Chinese and English”?
* In Section 5.2: “Frechet Inception Distances and Inception Scores” → “Frechet Inception Distances (FID) and Inception Scores (IS)”
* In Section 6: clarify “third-level super-resolution)
* In Section 7: “it is possible to train a classifier to distinguish the real and CogView2-generated images according to the texture features” is not supported with any evidence


**Limitations:**

I think some scoping about text is necessary. The paper generally assumes that input text can be any text in English and Chinese; their github repository explicitly says “any text” (https://github.com/THUDM/CogView2). For preciseness, it would be helpful to note any potential/practical limitations more clearly. For instance, CogLM can accept up to 111 text tokens (Section 3.2). And presumably, there aren’t that many short text inputs (e.g. one or two words) in the training data – then what would be a reasonable minimum length for the text input for the model to perform well? More generally, based on the types of images and texts in the training data, CogLM may perform better for certain kinds of images and texts. This insight can greatly help the future use of this pretrained model.

This applies to images as well. The fact that this paper only considers square images (based on N^2 notation) is not explicitly addressed.


**Strengths And Weaknesses:**

Strengths
* The paper is clearly written and aided with great visualizations. Overall, the problems and proposed solutions are well-motivated and supported with appropriate evidence or justification (e.g. findings from existing literature, their own empirical observations, or limitations due to compute resources).
* The topic is timely; the effort for making the task of text-to-image generation faster and better is of great interest to the community.
* Their approach of generating a low-resolution image and refining it to be a high-resolution image is simple and straightforward. The use of various techniques is adequately justified (e.g. masking strategy and attention mask) and ablated (e.g. clustering sampling and attention upweighting).
* “Faster” text-to-image generation:  One of the main contributions of this paper is to make text-to-image generation faster. Although the experiment section doesn’t directly compare different models’ inference time in an end-to-end fashion, the last paragraph of the introduction section mentions that their model run time for local parallel autoregressive generation is 600x faster and overall 10x faster than their previous model.
* “Better” text-to-image generation: The experiments follow popular benchmarking practice and discuss the gap between automatic metrics and human evaluation. With automatic metrics, CogView2 performs comparable to other methods on MS-COCO based on Frechet Inception Distances. Based on human evaluation, CogView2 performs better on all metrics (image clarity, texture quality, and relevance to the caption) than CogView, Lafite, and DF-GAN.

Weaknesses
* “Better” text-to-image generation: As the authors acknowledged in the paper, the automatic metrics on MS-COCO may not be the best way to evaluate these models, hence the claim of CogView2 being “better” (in the title) or competitive (in the abstract) compared to other models such as DALLE2 can use some scoping/hedges. Outside of this paper, there have been some informal qualitative comparisons between several models, which seem to give the impression that CogView2 is not strictly better than other models.
  * https://huggingface.co/spaces/THUDM/CogView2
  * https://twitter.com/bhagatsurya2/status/1542824988092530689
  * https://www.reddit.com/r/MachineLearning/comments/vkvq0j/r_cogview2_faster_and_better_texttoimage/
* Bilinguality: Since this paper considers text in both English and Chinese and notes that “Chinese input produces better results than English input” (https://huggingface.co/spaces/THUDM/CogView2), it would be interesting to see more in-depth analysis on this bilingual aspect of the model. The authors state that they used [BOE] to denote the beginning of English text and [BOC] for Chinese text, but do not justify the decision or discuss any findings based on two languages.

---

> ### Author Response · Authors · 2022-08-02
> **Response**
>
>
> Thank you very much for your careful and insightful review. We will explain your questions and concerns point by point.
> > “Better” text-to-image generation: As the authors acknowledged in the paper, the automatic metrics on MS-COCO may not be the best way to evaluate these models, hence the claim of CogView2 being “better” (in the title) or competitive (in the abstract) compared to other models such as DALLE2 can use some scoping/hedges. Outside of this paper, there have been some informal qualitative comparisons between several models, which seem to give the impression that CogView2 is not strictly better than other models.
> I think some scoping about text is necessary.
>
> Thank you for your advice. The ``better'' mainly compares CogView2 with CogView, and we will modify the abstract to lower the description in the revised version. We will also add some scoping in the description of the codes.
>
> > It would be interesting to see more in-depth analysis on this bilingual aspect of the model.
>
> Thank you for your suggestions. It would be interesting to look into this. Currently, we have some observation on this topic.
> (1) It is possible some words only appear in Chinese or English data.
> (2) we find that the multi-mapping between languages could attribute for a large part of the performance gap. For example, “一个提手提箱的人的照片” can be translated into English as "a photo of a person with a suitcase" or "a photo of human and suitcase" (not very native). We find that the former translation performs much better because "human" is not usually used in these cases, even though both "human" and "person" are "人" in Chinese. We anticipate that using a pretrained NLP model for deep text understanding, as in the recent work Imagen, can largely solve this problem.
>
> > What were the main challenges/blockers for directly comparing different models’ inference time in an end-to-end fashion?
>
> To compare with the specific previous text2image models is quite hard, because the speed is relevant to the model size, deployment efforts (e.g., TensorRT) and the machines. To compare LoPAR with AR-based models under the same conditions, we add a table of the time and FLOPs for a 4,096 sequence on an A100-40GB GPU with different AR-based methods. The model configurations are the same as the pretrained CogLM 6B. All AR-based methods, e.g. DALL-E, CogView, Make-A-Scene, et al. should share the same trends with the table.
>
> |     | FLOPs  | time | Memory (inference) |
> |  ----  | ----  | - | - |
> |Forward (also teacher forcing training) | $1.17*10^{14}$| 858 ms | 5,041MB
> |Autoregressive generation (No cache)| $4.81*10^{17}(4095\times)$ | about 1h | 5,041MB
> |**Autoregressive generation (cached)** | $1.17*10^{14}(1\times)$ | 225.9s | 4,865MB
> |LoPAR| $7.02*10^{14}(6\times)$ | 4.89s | 5,041MB
> |LoPAR+local_attention | $5.82*10^{14}$ | 3.41s | 352MB
>
> As discussed in the second paragraph (Line 13-17), our motivation is to increase the degree of parallelism for acceleration, even with more FLOPs. Autoregressive generation with cached hidden states have the same FLOPs with a teacher-forcing forward step, but is much slower (858ms vs 225.9s). We draw a diagram (https://i.imgur.com/T0Y9io2.png) for better understanding this. For LoPAR, it is exactly N (N=6 in our setting) times and FLOPs of forward steps.
>
> The results show that we reduce the time for generation from 225.9s to 3.41s (not including the first hierarchy of pure AR) and the memory from 5.4GB to 352MB. We will add the above part to a revised version.
>
> > Discussion on failure modes: Even from the cherry picked examples in Figure 1, there are multiple failure modes observed (e.g. different numbers or lengths of fingers). What are the main types of failure modes the authors or human annotators observed? Any difference when it’s in English vs. Chinese?
>
> In our observation, the main failure modes include:
> (1) Artifacts on repeated patterns. We find these problems usually happen on grass in the DSR step. Since this seldom happens for other objects, an assumption is that it is data-related. There is no enough high-resolution grass images in our datasets.
> (2) Details on eyes and fingers. There are some inconsistency on eyes, and sometimes the number of fingers are wrong. We find similar artifacts in DALL-E2 examples but no in Imagen. We hypothesize the reason is that the iterative steps are not enough.
> (3) Position bias. The objects or humans are more likely to collapse than the middle ones.
>
> We didn't find obvious differences about the failure modes for English or Chinese texts.
>
> > Nitpicking
>
> Thank you very much for your carefully reviewing and valuable suggestions. We will update them in a revised version.

---

> > ### Comment · Reviewer_swZ7 · 2022-08-09
> > **Thanks!**
> >
> > Thanks for the clarification and additional information!

---

### Official Review · Reviewer_86fW · 2022-07-12

**Rating:** 6
**Confidence:** 3
**Soundness:** 3 good
**Presentation:** 3 good
**Contribution:** 3 good

**Summary:**

Auto-regressive transformer-based text-to-image model can generate elegant pictures, but also face low-generation speed problems for high-resolution image generation. This is due to the extremely long sequence length and the auto-regressive decoding schema. In this work, the authors propose Cogview2, which is a hierarchical transformer and adopts local parallel auto-regressive generation instead of the global AR. The proposed Cogview2 achieves comparable or even better results than Cogview and speeds up the inference a lot. Overall, Cogview2 is an efficient version of Cogview but more fast benefiting from the local parallel decoding.

**Questions:**


-   Have you tried other local parallel generation manners? e.g. within a local patch, left to right or top to bottom generation in a non-autoregressive fashion?
-   The attention mask shown in Figure2 is somehow difficult to understand. Do you mean the token in green box is masked and going to be generated? This slightly confuses me.
-   What about the speed of iterative super-resolution compared with stacked image super-resolution models in Imagen?

**Ethics Review Area:**

["I don’t know"]

**Limitations:**

There is no limitation discussed in the current version. The authors have addressed the potential negative societal impact.

**Strengths And Weaknesses:**

## Strength
-   The proposed Cogview2 consists of several modules. First, they use the CogLM to generate a preliminary image with 20 $\times$ 20 tokens. The following super-resolution is based on this initial image, and I think a useful finding here is that we do not need to predict an image with a high resolution from the scratch, since a small 20 $\times$ 20 tokens may involve the information given the text. This may motivate the following researcher to design more efficient text-to-image models.
-   Local parallel autoregressive decoding is interesting. Traditional AR for text-to-image generation flatten the image into a 1-D sequence and generate the visual tokens from left to right. It is quite time-consuming and also ignores the spatial correlations.
-   Thanks to the flexibility of pretrained general language model, Cogview2 can perform image completion naturally
-   Achieve comparable results with auto-regressive modeling while speeding up the inference quite a lot.

## Weakness
There is no major concerns for me. Maybe, I would like to see a totally non-autoregressive model which can deliver comparable results with AR ones.

---

> ### Author Response · Authors · 2022-08-02
> **Response**
>
>
>
> Thank you very much for your valuable review. We will answer your questions point by point.
> > Have you tried other local parallel generation manners? e.g. within a local patch, left to right or top to bottom generation in a non-autoregressive fashion?
>
> Yes, these are methods we first tried but found that if adjacent tokens are generated at the same step, they were sometimes not coherent, because the two position are very relevant, but don't know each other during the step of generation.
>
> For example, we upsample a batch of unfiltered generated images of "一群穿着格子衬衫的程序员 (a group of programmers wearing plaid T-shirts)" using LoPAR and a top-to-bottom LoPAR (https://imgur.com/gallery/KYsMesa). We can see that the sleeves in subfigures 2 and 4 have more consistent patterns in diagonal style LoPAR.
>
> > The attention mask shown in Figure2 is somehow difficult to understand. Do you mean the token in green box is masked and going to be generated? This slightly confuses me.
>
> Sorry for this confusion. We will revise this part for a better introduction.
> The green part is actually not masked physically, but applied a casual mask (as in the right figure) so that they cannot be seen by tokens outside the green parts. We use the term "mask region" for the consistency with BERT/MAE. In this way, we can train infilling and token-by-token in the same way with very little modification on the sequence.
>
> > What about the speed of iterative super-resolution compared with stacked image super-resolution models in Imagen?
>
> Imagen has two levels of diffusion-based super-resolution. According to the paper, they use a DDIM cosine noise schedule (4,000 diffusion steps) for 64->256, and 1,000 diffusion steps for 256->1024. The number of steps means that we need to forward the model that times (4,000 and 1,000) during generation naively. From the paper, it is not clear whether they use any fast generation methods. In contrast, CogView2 need 6 forward steps for super-resolution.

---

### Official Review · Reviewer_1wZ4 · 2022-07-12

**Rating:** 3
**Confidence:** 5
**Soundness:** 2 fair
**Presentation:** 2 fair
**Contribution:** 2 fair

**Summary:**

This paper presents CogView2, which aims for faster and better text2image generation based on its previous work CogView. The key idea is to adopt the spirit of coarse to fine and generate low-resolution tokens first(20x20) and then do a super-resolution to 60x60 tokens for high-resolution images.

**Questions:**

Please refer to the weaknesses part.

**Limitations:**

Yes.

**Strengths And Weaknesses:**

Strengths:
1. This paper tries to solve two critical issues of the current auto-regressive-based text2image model: quality and efficiency.
2. This paper provides many visualization consequences in the paper.


Weaknesses:
1. This paper claims too many things but does not verify them clearly in experiments. a) The training and inference efficiency of text2image generation are not evaluated or compared with previous methods with wall clock time or FLOPS. 2) As introduced in the introduction section the CogLM is general for many tasks like infilling tasks, and image captioning, but these capabilities are not presented in the paper.

2. The comparison with current works is not convincing. For example, CovViiew2 didn't show significant improvements over previous methods on FID-0. Previous SOTA methods like Make-A-Scene and DALL-E2 did not report the results of FID-1 to FID-8 for comparison. Besides, some latest works like latent space diffusion and VQ-Diffusion are missed in the table for comparison.

3. The quality and fidelity of these generated samples presented in the paper are not that impressive. First, the generated image is still blurry. Second, we can observe clear unreasonable structures for the human hands or faces.

---

> ### Author Response · Authors · 2022-08-02
> **Response (about efficiency evaluation)**
>
> Thank you for your feedback, and we will address your concerns point by point as follows:
> > The training and inference efficiency of text2image generation are not evaluated or compared with previous methods with wall clock time or FLOPS.
>
> Thank you greatly for the valuable suggestion to add a clear table for FLOPs and time. In the submitted verison, we report the final achievement (10x faster than CogView) in Line 57. Here we present the time and FLOPs for a 4,096 sequence on an A100-40GB GPU with different AR-based methods. The model configs are the same as the pretrained CogLM 6B.
>
> |     | FLOPs  | time | Memory (inference) |
> |  ----  | ----  | - | - |
> |Forward (also teacher forcing training) | $1.17*10^{14}$| 858 ms | 5,041MB
> |Autoregressive generation (No cache)| $4.81*10^{17}(4095\times)$ | about 1h | 5,041MB
> |**Autoregressive generation (cached)** | $1.17*10^{14}(1\times)$ | 225.9s | 4,865MB
> |LoPAR| $7.02*10^{14}(6\times)$ | 4.89s | 5,041MB
> |LoPAR+local_attention | $5.82*10^{14}$ | 3.41s | 352MB
>
> As discussed in the second paragraph (Line 13-17), our motivation is to increase the degree of parallelism for acceleration, even with more FLOPs. Autoregressive generation with cached hidden states have the same FLOPs with a teacher-forcing forward step, but is much slower (858ms vs 225.9s). We draw a diagram (https://i.imgur.com/T0Y9io2.png) for better understanding this. For LoPAR, it is exactly N (N=6 in our setting) times and FLOPs of forward steps.
>
> The results show that we reduce the time for generation from 225.9s to 3.41s and the memory from 5.4GB to 352MB. We will add this
>  part to a revised version.
>
>
> To compare with the specific previous text2image models is quite hard, because the speed is relevant to the model size. All AR-based methods, e.g. DALL-E[1], CogView[2], Make-A-Scene[3], et al. should share the same trends with the above table.
> The diffusion-based methods have a totally different framework, whose FLOPs itself is $N\times$ that of forward, where $N$ is the diffusion steps (usually 1,000). The most prevalent acceleration is to make a trade-off between quality and speed using spaced sampling [4] or Analytic DPM [5]. They are not appropriate to be directly compared with AR-based methods.
>
> [1] Ramesh, Aditya, et al. "Zero-shot text-to-image generation." International Conference on Machine Learning. PMLR, 2021.
>
> [2] Ding, Ming, et al. "Cogview: Mastering text-to-image generation via transformers." Advances in Neural Information Processing Systems 34 (2021): 19822-19835.
>
> [3] Gafni, Oran, et al. "Make-a-scene: Scene-based text-to-image generation with human priors." arXiv preprint arXiv:2203.13131 (2022).
>
> [4] Nichol, Alex, et al. "Glide: Towards photorealistic image generation and editing with text-guided diffusion models." arXiv preprint arXiv:2112.10741 (2021).
>
> [5] Bao, Fan, et al. "Analytic-dpm: an analytic estimate of the optimal reverse variance in diffusion probabilistic models." arXiv preprint arXiv:2201.06503 (2022).
>
> If our answer and additional statistics above ease your concerns, could you increase your rating a bit? Please tell us if you have further concerns.

---

> ### Author Response · Authors · 2022-08-02
> **Response (about others)**
>
> > As introduced in the introduction section the CogLM is general for many tasks like infilling tasks, and image captioning, but these capabilities are not presented in the paper.
>
> Please see Appendix B for some results of the text-guided infilling task, and we use the image caption for post-selection as in CogView. We didn't report details about the image captions because it is not the focus of our work.
>
> > The comparison with current works is not convincing. For example, CovViiew2 didn't show significant improvements over previous methods on FID-0. Previous SOTA methods like Make-A-Scene and DALL-E2 did not report the results of FID-1 to FID-8 for comparison.
>
> First, we didn't claim CogView2 achieve better performance than DALL-E2, while instead we analyze the difference in section 6.
>
> Secondly, as we stressed in Line 270, **we need to downsample the images back to 256*256** for a meaningful FID comparison, which largely reduces the usage of our super-resolution method.
>
> Thirdly, FID itself is not a stable metric. According to https://www.cs.cmu.edu/~clean-fid/, even jpeg quality 75/100 can create an up to 20 FID difference. We also find whether center-crop COCO images create a >4 FID difference on this benchmark. We care more about human evaluation performance, where CogView2 outperforms CogView, LAFITE et al. by a large margin. However, many text-to-image models are not open-source, so that we cannot include them in the evaluation. This also suggests the value of open-sourcing of CogView2.
>
> > Besides, some latest works like latent space diffusion and VQ-Diffusion are missed in the table for comparison.
>
> Latent space diffusion first appeared as an unconditional generation paper, and updated a text-to-image model at the same time of our paper. Thank you for bringing it back to our scope. We will compare it in a revised version. We already cited VQ-Diffusion and will add it to the table. These methods are diffusion-based and not aim to generate high-resolution images.
>
> > The quality and fidelity of these generated samples presented in the paper are not that impressive. First, the generated image is still blurry. Second, we can observe clear unreasonable structures for the human hands or faces.
>
> The area is indeed developing very fast, and the recent DALL-E2, Imagen (after submission) and Parti (after submission) show better quality. However, The current text-to-image model is a large project, the final performance depends on many things, e.g. data, framework, resolution, parameters, et al. Our paper gives a concrete solution for a certain aspect -- the generation of high-resolution autoregressive models. In our opinion, this should also be encouraged. We discussed the way to improve our model in section 6, and the lack of deep text understanding revealed in Imagen might be the main reason of the gap, which is orthogonal to the contribution in this paper.

---

### Meta-Review · Area_Chair_xLRF · 2022-08-30

**Recommendation:** Accept
**Confidence:** Less certain

**Metareview:**

This paper describes a less auto-regressive approach for text-to-image generation. Reviewers were somewhat split, with one reject and one strong accept. Overall, the results the authors get are perhaps less compelling given the progress the field has made over the past few months since submission time, but I think the focus on less autoregression for this problem is important and potentially useful for other approaches to this problem, and also the authors had far fewer computational resources and smaller datasets than some of the other recent work, which makes me feel there is a chance the less-auto-regressive generation they employ will be useful for other text-to-image generation projects. I'm slightly less certain about this one because the results don't seem as compelling given works published after the submission deadline, and also the lack of experiments to investigate that this approach can be used with other text-2-image generation works. Still, it seems like an important direction that could use more papers, so I think this should be accepted. The bilingual generation is a bonus.

**Award:**

No

---

### Decision · Program_Chairs · 2022-09-14

Accept